# Optimal Configuration of Battery Energy Storage for AC/DC Hybrid System Based on Improved Power Flow Exceeding Risk Index

**Yanming Tu [1], Libo Jiang [2], Bo Zhou [1], Xinwei Sun [1], Tianwen Zheng [2,\*], Yunyang Xu [1] and Shengwei Mei [2]**

[1] State Grid Sichuan Electric Power Research Institute, Chengdu 610041, China; zbv_s@126.com (B.Z.); sunxiw09@126.com (X.S.)

[2] Sichuan Energy Internet Research Institute, Tsinghua University, Chengdu 610200, China; jliboc3@163.com (L.J.); Msw9821@126.com (S.M.)

\* Correspondence: tianwenscu@163.com

**Abstract:** After the fault disturbance (DC bi-polar blocking) in the AC/DC hybrid system, when the battery energy storage system (BESS) near the fault location is used to eliminate the power transfer, some sensitive and vulnerable transmission lines still have the problem of power flow exceeding the limit value. Therefore, an optimal configuration of BESS for AC/DC hybrid systems based on power flow exceeding risk index is proposed, which is used to eliminate the impact of power transfer on transmission lines. Firstly, considering the line outage distribution factor, the power flow exceeding risk index is established, which is used to judge the sensitive and vulnerable transmission lines on the shortest path power flow after the fault in the AC/DC hybrid system. The shortest path power flow is found by using the Dijkstra algorithm; the transmission lines nodes of the shortest path power flow are selected as candidate nodes for BESS configuration. Secondly, considering the safe and stable operation capability of the transmission lines, a multi-objective optimal mathematical model of BESS configuration for the AC/DC hybrid system is established, which minimizes the annual investment cost of BESS and maximizes the sum of the power flow exceeding risk index. Finally, the CEPRI36V7 power grid model in Power System Analysis Software Package (PSASP) is used for simulation analysis to verify the effectiveness of the proposed method.

**Keywords:** AC/DC hybrid system; battery energy storage system (BESS); improved power flow exceeding risk index; fault disturbance; optimal configuration; line outage distribution factor

## 1. Introduction

With the emergence of new phenomena, such as the widespread interconnection of power grids and the high penetration of renewable energy, in the past decades, cascading failures of power systems have caused several large-scale power outages worldwide, such as the power outages in California [1], and the disconnection accident in the power grid of Europe [2], which have caused huge economic losses and threatened the stable operation of power grids.

Due to its good technical and economic benefits in large capacity, long-distance, and flexible transmission, DC transmission technology has been widely used in long-distance power transmission, power grid interconnection, and other aspects. DC transmission technology has improved the ability of friendly large-scale renewable energy and effectively solved the imbalance between regional supply and demand of electricity [3]. With the continuous development of HVDC transmission technology and the application of many DC projects, China has built a large-scale complex AC/DC hybrid power grid [4,5]. However, the DC blocking fault in the DC transmission converter station will inevitably transfer the power from the DC transmission line to the AC transmission line, which will cause the power flow of the AC transmission line to exceed the thermal stability limit power and the

cascading failure. And then blackout accidents of power systems will happen, which seriously influence the local society and economy [6,7]. Therefore, to ensure the safe and stable operation of the AC/DC hybrid system, it is of great practical significance to eliminate the out-of-limit power flow caused by DC power transfer in AC transmission lines.

The energy storage has good dynamic active and reactive power regulation capabilities, and it can adapt to operational control requirements of different time scales. To reduce the load shedding after the system failure, improve the system operation flexibility and stability, and ensures the safe, reliable, and efficient operation of the AC/DC hybrid system, the DC power transfer of the AC/DC hybrid system is eliminated by using the energy storage [8]. However, due to the different impact of DC power transfer on other lines, the energy storage near the fault location is used to eliminate the power transfer after the fault disturbance (DC bi-polar blocking) in the AC/DC hybrid system, some sensitive and vulnerable transmission lines still have the problem of power flow exceeding the limit value. It is necessary to quickly identify the sensitive line set that has a great impact on the transmission power, and configure the energy storage in this node line, which can quickly eliminate the power limit, improve the system stability, and prevent the occurrence of major accidents.

At this stage, the method of identifying the vulnerability of the power system based on the dynamic characteristics of the power grid has been widely used [9]. In [10], the risk theory assessment method is used to identify key lines by simulating the hidden fault model in the chain fault, but it requires many simulation results to determine the probability of line disconnection through tests, which increases the workload and is difficult to achieve online application and reduces the practicality of the project. In [11], a vulnerability assessment method of power grid cascading fault propagation elements based on power flow entropy is proposed, which can distinguish the vulnerability of branches from impact and consequence. This method accurately models the physical characteristics of the power grid and can improve the simulation speed by reducing the fault search space. However, there is still a contradiction between sampling times and simulation accuracy, which is difficult to achieve online application. In [12], a comprehensive index is proposed to identify the vulnerable lines, which applies the impact vulnerability to represent its impact-resistance ability and the transfer vulnerability to represent the damage caused by its removal from the system. However, this method does not consider the margin of power flow out-of-limit capacity of other lines after the disconnected transmission line. The method of identifying the sensitive and vulnerable transmission lines based on power flow exceeding risk index is proposed in [13]. The method would not have to repeatedly calculate the impedance matrix of the line disconnection and connection. But it is difficult to apply to AC/DC hybrid systems, weak power grids, and other power systems. The scope of the application is limited. The improved power flow exceeding risk index is used for AC/DC hybrid systems and other various power systems. And this method preserved the advantages of traditional methods.

The optimal configuration of BESS is mainly to determine its optimal access location and capacity, to better play its performance, and to improve the absorption rate of renewable energy. In [14,15], the proposed coordinated operational planning for wind farms with BESS is that it can reduce the impacts of wind power forecast errors. Considering the uncertainty and curtailment rate constraint of wind power, reference [16] focuses on the BESS configuration method in wind farms. In [17], the capacity allocation of BESS is used to smooth wind power fluctuations, and the BESS capacity size at different confidence levels is studied. In [18], this paper proposes a bi-level optimal energy storage system (ESS) siting and sizing algorithm to mitigate the voltage deviation in distribution networks. A capacity allocation method of BESS in secondary frequency regulation with the goal of maximum net benefit is proposed in [19]. The literature [14–19] focuses on a single application scenario, such as reducing prediction error, improving new energy consumption, and ensuring power grid stability to achieve BESS configuration, which has significant limitations. They do not fully explore their advantages in coordinated operation or multiple application

scenarios. Large BESS capacity needs to be configured, and the utilization rate of BESS is low.

References [20–22] propose an optimized configuration method for the coordinated operation of BESS and renewable energy. Collaborative configuration of distributed generation and BESS in microgrids considering the state of health is studied in [23]. In [24], considering the uncertainty of the net load, this study provides an approach to analyzing the BESS demand capacity for peak shaving and frequency regulation. In reference [25], the feasibility and compatibility of using such idle capacity and power of BESS to participate in the electricity energy market and reserve ancillary service market are explored, and a coordinated operation strategy for the three application scenarios of BESS is proposed to improve its utilization. When the power system is in a steady state, to achieve economic efficiency, ensure grid stability, and improve the utilization rate of BESS, the optimization configuration of BESS for the collaborative operation of BESS and renewable energy and multiple application scenarios of BESS services is studied by domestic and foreign scholars. But the frequent occurrence of extreme weather would seriously affect the safe and stable operation ability of the power grid. It is necessary to study the optimal configuration of BESS considering the influence of extreme weather on the power grid. This can enhance the safe and stable operation capacity of the power grid.

To cope with the impact of extreme weather, such as typhoons and freezing rain, on the power grid, BESS has been configured to improve the reliability and flexibility of the power grid in recent years. Literature [26] takes the load-shedding cost of the system under extreme events as the toughness index and studies the optimal allocation of BESS considering the toughness of the distribution network. In [27], a distributed energy storage planning model for the distribution network considering the influence of typhoon weather is established, and a decomposition collaborative solution method based on the Benders decomposition is proposed. In [28], a new quantitative index of toughness and formulates of a method of BESS planning were proposed to enhance the seismic capacity of the distribution network. Literature [29] proposed a distribution network BESS planning method considering toughness and established a two-stage robust optimization model, which can effectively ensure the uninterrupted power supply of important loads. The above literature configures BESS to improve the flexibility or toughness of the power grid by ensuring a continuous power supply of important loads in extreme weather. However, with the increase in the penetration rate of new energy, the probability of power grid failures has increased, such as exceeding the power limit of transmission lines and cascading faults in the power grid. It is very necessary to allocate BESS reasonably after a power grid failure, such as to quickly eliminate the over-limit of AC line power and improve the stability of the system; how to reasonably configure BESS after the DC locking fault occurs in the AC/DC hybrid system.

And then, a large amount of research has been conducted domestically and internationally on solution methods for BESS optimization configuration. Intelligent optimization algorithms, such as genetic algorithm [30] and particle swarm optimization (PSO) [31], have been widely applied. The optimization configuration method proposed provides a good reference for the solution in this article.

Aiming at the advantages and disadvantages of the existing research, considering the millisecond level active dynamic response capability of the BESS system, an optimized configuration of BESS in the AC/DC hybrid system based on the improved power flow exceeding risk index is proposed. Firstly, the improved power flow exceeding risk index is established to evaluate the sensitivity and vulnerability of other lines to the transferred power flow after the branch is disconnected. Secondly, the Dijkstra algorithm is used to find out the shortest path of the closed loop formed by the breaking line, and the key nodes are selected as the candidate sites for BESS by calculating the improved power flow exceeding risk index of the shortest path. Finally, a multi-objective function with the maximum sum of improved power flow exceeding risk index and the minimum annual investment cost of BESS is established, and particle swarm optimization (PSO) is used to obtain the optimal

configuration scheme of BESS in the AC/DC hybrid system. Meanwhile, the BESS system adopted the active power control strategy, including plant-level control and local control, which quickly eliminates the power exceeding the limit of the AC line and suppresses the power fluctuation of the power grid.

This paper is organized as follows: The identification of sensitive and vulnerable lines is talked about in Section 2. The mathematical model for the optimal allocation of BESS is proposed in Section 3. The Model-solving method is given in Section 4. In Section 5, the effectiveness and feasibility performance of the proposed method are examined on the CEPRI36V7 grid model. Section 6 is the conclusion.

## 2. Identification of Sensitive and Vulnerable Lines

### 2.1. Line Outage Distribution Factor

If line A of the AC/DC hybrid system is faulty, and it causes the line disconnection (DC line causes blocking fault, etc.), which causes the transfer of active power flow in the system, that is, the active power flow of other lines is changed. The relationship between the change of normal line power flow and the original power flow of the disconnected line can be expressed by the Line Outage Distribution Factor (LODF) [32]:

$$D_{R-A} = \frac{\Delta P_{R-A}}{P_A} \tag{1}$$

where $\Delta P_{R-A}$ is the change of line $R$'s active power flow after line $A$ is disconnected; $D_{R-A}$ is the LODF that causes the change of line $R$'s active power flow after line $A$ is disconnected; $P_A$ is the steady-state initial active power of line $A$.

Assuming that the nodes at both ends of line $A$ are $i$ and $j$, and the injected active power remains unchanged before and after disconnection, the change of node active power flow caused by line $A$ disconnection is

$$\Delta P = [0 \cdots 1 \cdots -1 \cdots 0]^T P_A = M_A P_A \tag{2}$$

where $M_A$ is the node-branch associated $n \times 1$ order column vector of branch $A$, and the row corresponds to the node number.

The $n \times 1$ order change $\Delta \theta$ of node voltage phase angle caused by line $A$ disconnection. $\Delta \theta$ can be expressed as

$$\Delta \theta = (B - M_A x_A^{-1} M_A^T)^{-1} M_A P_A \tag{3}$$

where $B$ is the $n \times n$ order admittance matrix; the admittance matrix is sparse type; $x_A$ is the reactance of line $A$.

Then, the change of active power flow of branch $R$ ($R \neq A$) caused by line $A$ disconnection is

$$\Delta P_{R-A} = \frac{M_R^T \Delta \theta}{x_R} = D_{R-A} P_A = \frac{M_R^T (B - M_A x_A^{-1} M_A^T)^{-1} M_A P_A}{x_R} \tag{4}$$

where $M_R$ is the node-branch associated $n \times 1$ order column vector of branch $R$; $x_R$ is the 1 order reactance of line $R$.

Let $B^{-1} = X$, after simplification, the expression of $D_{R-A}$ is:

$$D_{R-A} = \frac{X_{R-A}/x_R}{1 - X_{A-A}/x_R} \tag{5}$$

Among them,

$$X_{R-A} = M_R^T X M_A \tag{6}$$

$$X_{A-A} = M_A^T X M_A \tag{7}$$

where $X$ is the $n \times n$ order impedance matrix; $X_{A-A}$, $X_{R-A}$ is the 1 order self-impedance and mutual impedance between nodes of port $R$ and port $A$, respectively.

After the DC blocking fault occurs in the AC/DC hybrid system, the influence of DC power transfer on the AC line can be measured by calculating the LODF of the AC line.

*2.2. The Improved Power Flow Exceeding Risk Index*

The improved power flow exceeding risk index takes into account the impact of power transfer on other lines and the margin of the out-of-limit capacity of line power flow. After considering the margin of the out-of-limit capacity of line power flow, it is not necessary to consider the problem of reverse power flow of other lines caused by power flow transfer separately, which reduces unnecessary calculations, and can better reflect the sensitivity and vulnerability of other lines, and identify the sensitive and vulnerable lines in the system. Under the change of power flow of line R caused by the disconnection of line A, the margin of the out-of-limit capacity of the power flow of line R can be expressed as follows:

When $D_{R-A} < 0$, this is true, the expression of line power flow out-of-limit capacity margin is as follows:

$$\Delta P' = \begin{cases} |P_{R,\max} + P_R| & P_R \geq 0 \\ |-P_{R,\max} - P_R| & P_R < 0 \end{cases} \tag{8}$$

When $D_{R-A} > 0$, this is true, the expression of line power flow out-of-limit capacity margin is as follows:

$$\Delta P' = \begin{cases} |-P_{R,\max} + P_R| & P_R \geq 0 \\ |P_{R,\max} - P_R| & P_R < 0 \end{cases} \tag{9}$$

Combined with the LODF, the improved power flow exceeding risk index is given to evaluate the sensitivity and vulnerability of other lines to the transferred power flow after the branch break, as follows:

$$\Phi_{R-A} = \frac{\Delta P'}{D_{R-A} \cdot P_{A,\max}} \tag{10}$$

where $P_{A,\max}$ is the thermal stability limit value of the breaking line.

After a DC blocking fault occurs in the AC/DC hybrid system, the sensitivity and vulnerability of each AC line can be effectively evaluated by calculating the improved power flow exceeding risk index of each AC line. The smallest the improved power flow exceeding risk index of each AC line, the lower its ability to receive the transferred power flow, and the higher the improved power flow exceeding risk index. This paper selects the sensitive vulnerability line when the absolute value of the improved power flow exceeding risk index is less than 0.5. Secondly, after the DC blocking fault occurs in the AC/DC hybrid system, the DC power flow is mainly transferred to the shortest path that forms a closed loop with the DC line, so the shortest path set of DC power transfer needs to be searched.

*2.3. The Shortest Path Search Based on the Dijkstra Algorithm*

Using the knowledge of graph theory, the AC/DC hybrid system is simplified and abstracted into a graph G (V, E), where V represents the bus set in the grid, E represents the line set between buses, and the line side weight value is the line reactance. Then, the shortest path algorithm related to graph theory is adopted.

The shortest path search algorithms in graph theory include the Dijkstra algorithm, Floyd algorithm, etc. The Dijkstra algorithm has small time complexity and is easy to expand; the Floyd algorithm has high time complexity and space complexity, which increases the calculation amount. Its advantage is that it can be used to search the shortest path of the line with negative weight. Because of the scalability of the Dijkstra algorithm and the fact that there are no branches with negative weights in the graph, and to meet the requirements of fast calculation, the shortest path search based on the Dijkstra algorithm is adopted.

After the DC blocking fault occurs in the AC/DC hybrid system, the shortest path forming a closed loop with the DC line can be searched by using the Dijkstra algorithm, and then the sensitive AC line can be identified by combining the improved power flow exceeding risk index, and the power input node of the sensitive, vulnerable line is used as the candidate location for configuring BESS.

## 3. The Mathematical Model for Optimal Allocation of BESS

### 3.1. Objective Function

After the DC locking fault occurs in the AC/DC hybrid system, when the system optimizes the configuration of BESS, it should also have a certain economy while eliminating the power flow over-limit on the sensitive and vulnerable lines. Therefore, the multi-objective function of the optimal configuration of BESS in the AC/DC hybrid system is as follows:

$$\max \Gamma = \sum_{k=1}^{K} \left| \frac{\Delta P'}{D_{k-A} \cdot P_{\mathrm{dc-max}}} \right| \tag{11}$$

$$\min G_{inv} = \frac{r(1+r)^y}{(1+r)^y - 1} \cdot (c_1 \cdot P_e + c_2 \cdot E_e) \tag{12}$$

where $\Gamma$ is the sum of the improved power flow exceeding risk index of sensitive vulnerable lines after DC blocking fault; $D_{k-A}$ is the LODF; $P_{\mathrm{dc-max}}$ is the thermal stability limit value of DC line; $G_{inv}$ is the annual investment cost of BESS; $c_1, c_2$ are the unit power cost and capacity cost of BESS; $P_e, E_e$ are the rated power and rated capacity of the BESS, respectively; $r$ is the annual interest rate of the fund; $Y$ is the life cycle of BESS; $K$ is the sensitive vulnerability line.

### 3.2. Constraint

(1) Power balance constraints

$$\sum_{m=1}^{M} P_{G,m} + \sum_{n=1}^{N} P_{L,n} + \sum_{s=1}^{S} P_{B,s} = 0 \tag{13}$$

where $P_{G,m}$, $P_{L,n}$, $P_{B,s}$ are, respectively, the output of generator $m$, the required power of load $n$, and the charging and discharging power of BESS s; $S$ is the quantity of configured BESS in the system; $M$ and $N$ are the number of generators and the number of loads in the system.

(2) Line loss constraint

$$P'_R U_R^2 \geq R_R \cdot (P'^2_R + Q'^2_R) \tag{14}$$

where $P'_R$, $Q'_R$ are the active power and reactive power transmitted by the receiving end of the $R$ line, respectively; $R_R$ is the resistance of the $R$ line; $U_R$ is the voltage amplitude of the receiving terminal node of the Rth line.

(3) Generator power constraint

$$\overline{P}_{G,m} \leq P_{G,m} \leq \underline{P}_{G,m} \tag{15}$$

where $\overline{P}_{G,m}$, $\underline{P}_{G,m}$ are the upper and lower limits of generator output, respectively.

(4) Line power constraint

$$\overline{P}_{L,m} \leq P_{L,m} \leq \underline{P}_{L,m} \tag{16}$$

where $\overline{P}_{L,m}$, $\underline{P}_{L,m}$ are the upper and lower limits of the active power of the transmission line.

(5) Capacity constraints of BESS

$$S_{\min} \leq S_r \leq S_{\max} \tag{17}$$

where $S_{min}$, $S_{max}$ are the minimum and maximum capacities of BESS, respectively.

(6)　Power constraint of BESS charge and discharge

$$-P_{r,max} \leq P_r^c \leq 0$$
$$0 \leq P_r^d \leq P_{r,max} \tag{18}$$

where $P_{r,max}$ is the maximum value of BESS discharge power; $P_r^c$, $P_r^d$ are the charge power and discharge power of the BESS system, respectively.

## 4. Model-Solving Method

The sharing of information among the entire population is beneficial for the population towards a better position in genetic algorithms. Only the best individual's information is shared in PSO, and the entire search process is tracking the optimal solution. So, the PSO algorithm has faster convergence than the genetic algorithm. And due to its advantages of high accuracy and fast convergence, the PSO algorithm is widely used in BESS capacity configuration [33,34]. Therefore, this article chooses the PSO algorithm for solving BESS capacity configuration.

To reasonably obtain the location of BESS, the PSO algorithm is used to solve for the optimal capacity of BESS, and the optimal location of BESS is selected from candidate nodes. Meanwhile, BESS adopts the active power control strategy, including plant-level control and local control, which quickly eliminates the power exceeding the limit of the AC line, suppresses the power fluctuation of the power grid, and ensures the safe and stable operation ability of the power grid.

The specific process is shown in Figure 1, and the solution steps are as follows:

(1)　The graph obtained by abstracting the power grid is G0, and the reactance value of each line in the power system is taken as the weight value of each side;

(2)　After DC blocking occurs in the converter station at the sending end, the Dijkstra algorithm is used to find the shortest path between the converter station and the designated node;

(3)　The branch set contained in the target source point and destination point are combined to obtain the branch set of power flow transfer;

(4)　Calculate the improved power flow exceeding risk index of all branches in the power flow transfer branch set, and select the branches whose absolute value of the improved power flow exceeding risk index is less than 0.5 to form the main branch set;

(5)　For the lines in the main branch set, if there is a reverse flow and the branch flow meets $|P_R'| \leq |P_R|$, it will be removed from the main branch set ($P_R'$ is the branch flow after the line is disconnected);

(6)　Input parameters. Input PSO controlling variables of the original parameters. Set PSO algorithm parameters: the maximum iteration number is 300, and the population size is 200;

(7)　Initialize the population. According to Equation (19), the N solutions are generated, such as the energy storage power and capacity, and they also are guaranteed to satisfy the condition. The objective function value is calculated for all the scenes using Equations (11) and (12).

$$x_{ij} = x_j^{min} + rand() \cdot (x_j^{max} - x_j^{min}) \tag{19}$$

where $i = 1, 2, \ldots, N$ is a D-dimension vector; $j = 1, 2, \ldots, d$; $rand()$ represents random numbers between 1 and 0; $x_j^{max}$, $x_j^{min}$ are the maximum and minimum values of particles, respectively;

(8) Calculate the fitness value for particles by using (20). It is updated local optimal position and global optimal position by using Equations (21) and (22).

$$p_i = \frac{F_i}{\sum\limits_{k=1}^{N} F_k} \tag{20}$$

$$v_{ij}(t+1) = w \times v_{ij}(t) + c_1 \times rand() \times \left(p_{ij}(t) - x_{ij}(t)\right) + c_2 \times rand() \times \left(p_{gj}(t) - x_{ij}(t)\right) \tag{21}$$

$$x_{ij}(t+1) = x_{ij}(t) + v_{ij}(t+1) \tag{22}$$

where $F_i$ is the corresponding fitness value for particles I; $v_{ij}(t+1)$, $v_{ij}(t)$ are the velocity of the ith particle at $t+1$, $t$ times, respectively; $w$ is the inertia factor; $w = 0.8$, $c_1$ and $c_2$ are the learning rate; $c_1 = 0.9, c_2 = 0.9$. $p_{ij}(t)$, $p_{gj}(t)$ respectively represent the individual optimal value and the global optimal value of particles;

(9) Output optimal solution. If the iteration number is greater than the set value, then output the Parote optimal. Otherwise, return to step (7).

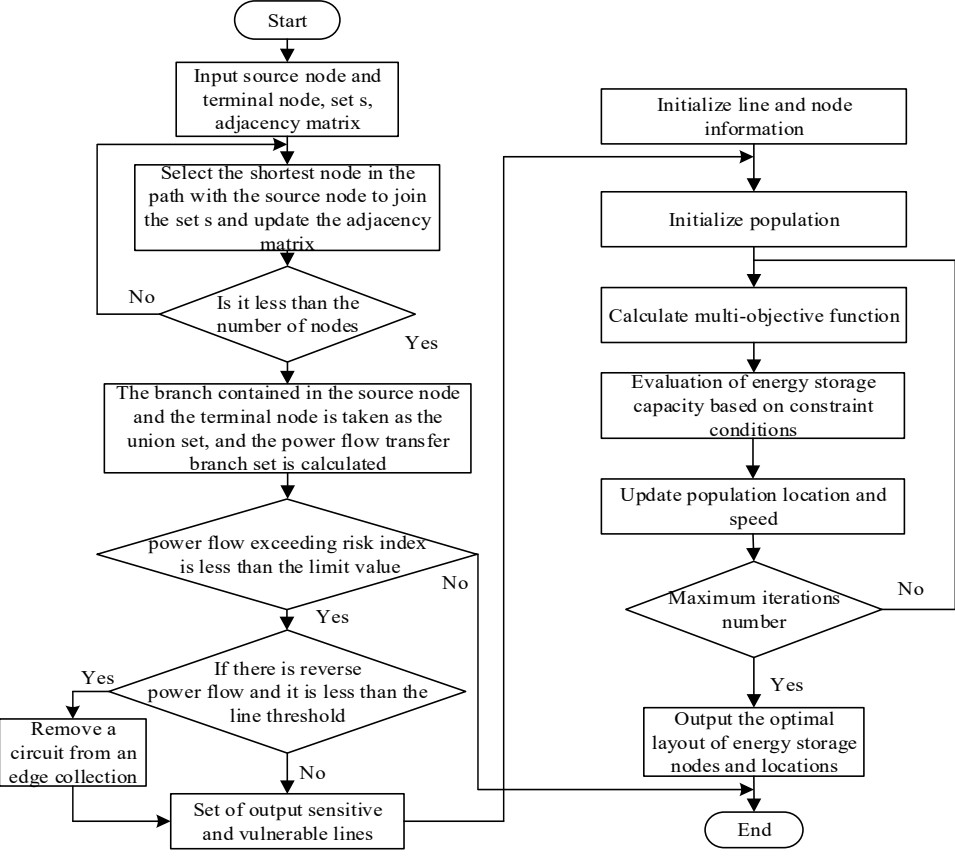

**Figure 1.** Solution flow of BESS optimization configuration.

## 5. Simulation Analysis

### 5.1. Parameter Design

The CEPRI36V7 power grid model in the power system analysis comprehensive program (PSASP) was used for simulation analysis to verify the effectiveness of the proposed BESS configuration strategy. The parameters of the CEPRI36V7 model are referred to in reference [35]. The topology of the CEPRI36V7 model is shown in Figure 2. The parameters of the CEPRI36V7 model are referred to in reference [35]. The node parameters, generator parameters, and branch parameters of the CEPRI36V7 model are shown in Tables 1–3, respectively.

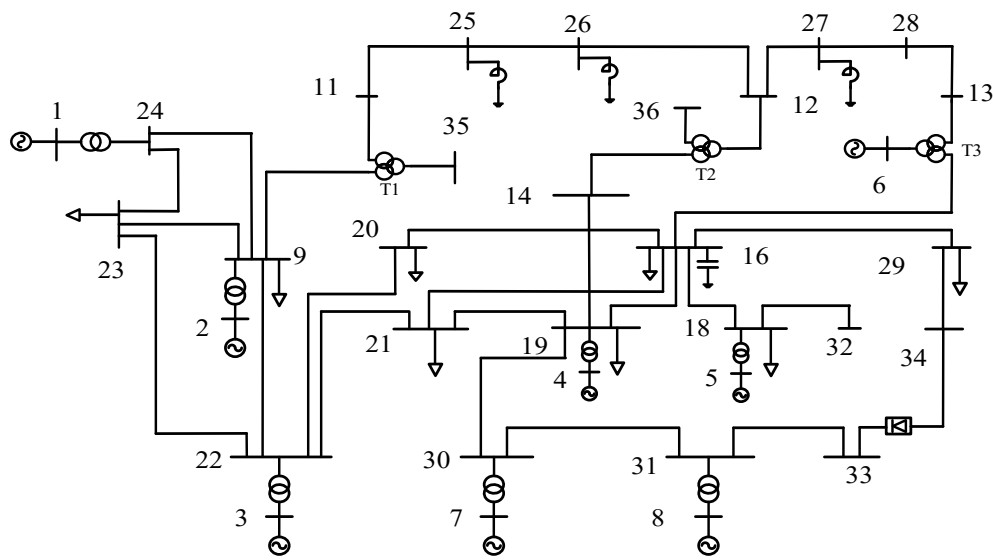

**Figure 2.** The power grid structure of CEPRI36V7.

**Table 1.** The node parameters of the CEPRI36V7 model.

| Bus_i | Type | $P_d$/MW | $Q_d$/Mvar | Base/kV | Bus_i | Type | $P_d$/MW | $Q_d$/Mvar | Base/kV |
|-------|------|----------|------------|---------|-------|------|----------|------------|---------|
| 1 | 3 | 0 | 0 | 10.5 | 19 | 1 | 86.4 | 66.2 | 220 |
| 2 | 1 | 0 | 0 | 20 | 20 | 1 | 71.9 | 47.4 | 220 |
| 3 | 2 | 0 | 0 | 10.5 | 21 | 1 | 70 | 50 | 220 |
| 4 | 1 | 0 | 0 | 15.7 | 22 | 1 | 226.5 | 169 | 220 |
| 5 | 1 | 0 | 0 | 10.5 | 23 | 1 | 287 | 144 | 220 |
| 6 | 2 | 0 | 0 | 10.5 | 24 | 1 | 0 | 0 | 220 |
| 7 | 2 | 0 | 0 | 10.5 | 25 | 1 | 0 | 0 | 500 |
| 8 | 2 | 0 | 0 | 10.5 | 26 | 1 | 0 | 0 | 500 |
| 9 | 1 | 376 | 221 | 220 | 27 | 1 | 0 | 0 | 500 |
| 10 | 1 | 0 | 0 | 20 | 28 | 1 | 0 | 0 | 500 |
| 11 | 1 | 0 | 0 | 500 | 29 | 1 | 520 | 10 | 220 |
| 12 | 1 | 0 | 0 | 500 | 30 | 1 | 0 | 0 | 220 |
| 13 | 1 | 0 | 0 | 500 | 31 | 1 | 0 | 0 | 220 |
| 14 | 1 | 0 | 0 | 220 | 32 | 1 | 0 | 0 | 220 |
| 15 | 1 | 0 | 0 | 20 | 33 | 1 | 0 | 0 | 220 |
| 16 | 1 | 500 | 230 | 220 | 34 | 1 | 0 | 0 | 220 |
| 17 | 1 | 0 | 0 | 20 | 35 | 1 | 0 | 0 | 0 |
| 18 | 1 | 430 | 220 | 220 | 36 | 1 | 0 | 0 | 0 |

**Table 2.** The generator parameters of the CEPRI36V7 model.

| Bus | $P_g$/MW | $Q_g$/Mvar | $V_g$/p.u. |
|---|---|---|---|
| 1 | 0 | 0 | 1 |
| 2 | 600 | 360 | 1 |
| 3 | 310 | 0 | 1 |
| 4 | 160 | 70 | 1 |
| 5 | 430 | 334 | 1 |
| 6 | −1 | 0 | 1 |
| 7 | 225 | 0 | 1 |
| 8 | 306 | 0 | 1 |

**Table 3.** The branch parameters of the CEPRI36V7 model.

| Fbus | Tbus | r | x | b | Ratio | Fbus | Tbus | r | x | b |
|---|---|---|---|---|---|---|---|---|---|---|
| 11 | 25 | 0 | 0.0001 | 0 | 0 | 31 | 32 | 0 | 0.0001 | 0 |
| 12 | 26 | 0 | 0.0001 | 0 | 0 | 9 | 22 | 0.0559 | 0.218 | 0.3908 |
| 12 | 27 | 0 | 0.0001 | 0 | 0 | 9 | 23 | 0.0034 | 0.0131 | 0 |
| 13 | 28 | 0 | 0.0001 | 0 | 0 | 9 | 24 | 0.0147 | 0.104 | 0 |
| 14 | 19 | 0.0034 | 0.02 | 0 | 0 | 24 | 1 | 0 | 0.015 | 0 |
| 16 | 18 | 0.0033 | 0.0333 | 0 | 0 | 9 | 2 | 0 | 0.0217 | 0 |
| 16 | 19 | 0.0578 | 0.218 | 0.3774 | 0 | 22 | 3 | 0 | 0.0124 | 0 |
| 16 | 20 | 0.0165 | 0.0662 | 0.4706 | 0 | 19 | 4 | 0 | 0.064 | 0 |
| 16 | 21 | 0.0374 | 0.178 | 0.328 | 0 | 18 | 5 | 0 | 0.0375 | 0 |
| 16 | 29 | 0 | 0.0001 | 0 | 0 | 30 | 7 | 0 | 0.0438 | 0 |
| 18 | 34 | 0 | 0.001 | 0 | 0 | 31 | 8 | 0 | 0.0328 | 0 |
| 19 | 21 | 0.0114 | 0.037 | 0 | 0 | 12 | 15 | 0 | 0.018 | 0 |
| 19 | 30 | 0.0196 | 0.0854 | 0.162 | 0 | 6 | 17 | 0 | 0.0337 | 0 |
| 20 | 22 | 0.0214 | 0.0859 | 0.6016 | 0 | 9 | 10 | 0 | −0.002 | 0 |
| 21 | 22 | 0.015 | 0.0607 | 0.4396 | 0 | 14 | 15 | 0 | −0.002 | 0 |
| 22 | 23 | 0.0537 | 0.19 | 0.3306 | 0 | 13 | 17 | 0 | 0.01 | 0 |
| 23 | 24 | 0.0106 | 0.074 | 0 | 0 | 11 | 10 | 0 | 0.018 | 0 |
| 25 | 26 | 0.0033 | 0.0343 | 3.7594 | 0 | 36 | 15 | 0 | 0.0001 | 0 |
| 27 | 28 | 0.00245 | 0.0255 | 2.79 | 0 | 16 | 17 | 0 | 0.001 | 0 |
| 29 | 33 | 0 | 0.0001 | 0 | 0 | 35 | 10 | 0 | 0.001 | 0 |
| 30 | 31 | 0 | 0.0001 | 0 | 0 | | | | | |

The capacity of the CEPRI36V7 power grid is 2600 MW, and nodes 33 to 34 are DC transmission lines, with a DC transmission capacity of 2 × 200 MW; DC power flows from 33 nodes to 34 nodes, and nodes 10, 15, and 17 in the system are the central nodes of three-winding transformers T1, T2, and T3.

This article uses lithium BESS. The maximum rated power and capacity of the total installed BESS are 800 MW and 1600 MW. h respectively, and the cost coefficients are 1500 yuan/kW and 2000 yuan/kWh. The charge and discharge rate of BESS is 0.5 C.

### 5.2. Sensitive Line Identification

The simplified structure of the power grid is shown in Figure 3. The red line represents the DC bipolar locking fault line. The blue line and black line represent the non-shortest path AC line. The green line represents the shortest path AC line. The converter station of node 33 has a DC bipolar locking fault, and the Dijkstra algorithm is used to obtain the shortest path composed of node 33 and node 34, namely: 33, 31, 30, 19, 14, 15, 12, 27, 28, 13, 17, 16, 29, and 34, a total of 13 AC lines. The power change curve of the AC line is shown in Figures 4 and 5 after the converter station of node 33 has a dual-machine locking fault at 5 s.

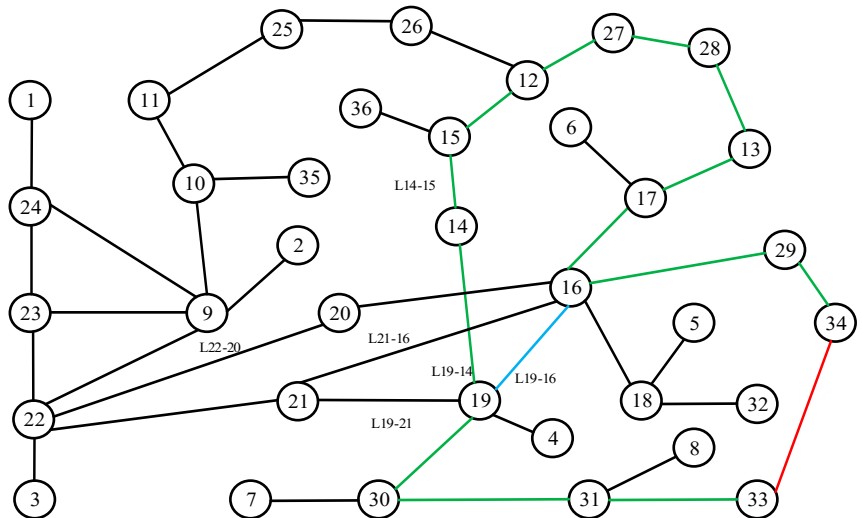

**Figure 3.** The simplified grid structure of CEPRI36V7.

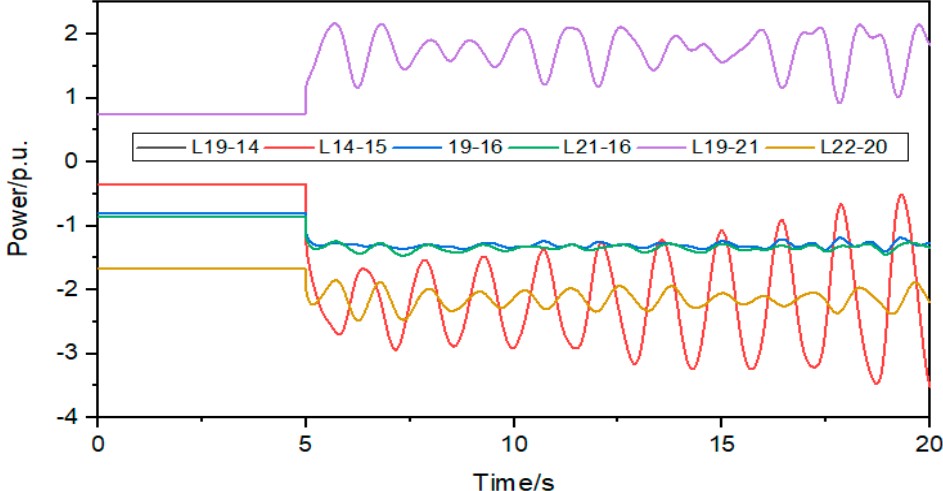

**Figure 4.** Change curve of AC line after DC blocking.

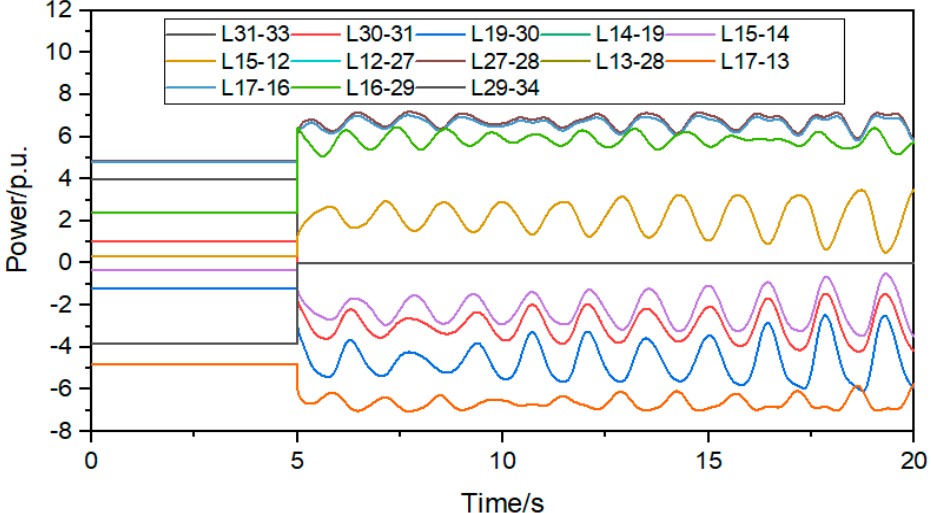

**Figure 5.** Power change curve of the shortest path line after DC blocking.

It can be seen from Figure 4 that the power variation of lines L19-14 and L14-15 is the largest, with a power variation greater than 2 p.u., and the power variation of other lines is less than 1.3 p.u. It can be seen from Figure 5 that the power variation of the AC line on the shortest path is greater than 1.5 p.u. It can be seen that after the DC bipolar locking fault, the power flow is mainly in the shortest path composed of node 33 and node 34.

The improved power flow exceeding risk index of each line on the shortest path is shown in Table 4 after the bipolar locking fault occurs at the converter station of node 33 at 5 s. From Table 4, it can be seen that the absolute value of the improved power flow exceeding risk index of lines L30-31, L19-30, L14-19, L15-14, L15-12, L17-13, L17-16, and L16-29 is less than 0.5, so the above AC lines are sensitive and vulnerable. Since node 15 and node 17 are the central nodes of the three-winding transformer, and BESS is configured at the sending end of the AC line, the candidate nodes for BESS are 31, 19, 14, and 16.

**Table 4.** The improved power flow exceeding risk index of the shortest path.

| AC Line | Initial Power/p.u. | Power after Fault/p.u. | LODF | The Improved Power Flow Exceeding Risk Index |
|---|---|---|---|---|
| L31-33 | 4.00 | 0.00 | −1.00 | 1.67 |
| L30-31 | 1.02 | −2.98 | −1.00 | 0.41 |
| L19-L30 | −1.20 | −5.17 | −1.00 | 0.70 |
| L14-19 | −0.34 | −2.94 | −0.82 | 0.42 |
| L15-14 | −0.34 | −2.94 | −0.65 | 0.32 |
| L15-12 | 0.34 | 2.94 | 0.65 | 0.14 |
| L12-27 | 4.86 | 7.18 | 0.58 | 1.29 |
| L27-28 | 4.86 | 7.18 | 0.58 | 1.29 |
| L13-28 | −4.61 | −7.04 | −0.61 | 3.70 |
| L17-13 | −4.80 | −7.04 | −0.56 | 3.22 |
| L17-16 | 4.79 | 7.01 | 0.56 | 0.36 |
| L16-29 | 2.38 | 6.20 | 1.02 | 0.32 |
| L29-L34 | −3.82 | 0.00 | −0.96 | 2.15 |

### 5.3. Optimization Configuration Results of Single BESS

To verify that the BESS is configured in the sensitive and vulnerable line, the BESS to improve the improved power flow exceeding risk index is the best. Firstly, the PSO algorithm is used to obtain a set of Pareto solutions of the BESS configuration capacity and location, as shown in Table 5.

**Table 5.** The single BESS configuration capacity and location.

| ESS Power/MW | ESS Capacity/MW·h | ESS Location | Annual Investment Cost/Million Yuan | the Sum of the Improved Power Flow Exceeding Risk Index |
|---|---|---|---|---|
| 541.15 | 541.15 × 2 | 31 | 443.47 | 24.18 |
| 276.31 | 276.31 × 2 | 16 | 226.44 | 18.29 |
| 391.88 | 391.88 × 2 | 16 | 321.15 | 19.47 |

From Table 5, there are two sets for the BESS configuration at node 16. There are two sets for the BESS configuration at node 16. The annual investment cost of BESS is 226.44 million yuan and 321.15 million yuan. Similarly, the BESS configuration also obtains the optimal solution at 31 nodes; the annual investment cost of BESS is 226.44 million yuan

and 443.47 million yuan. Therefore, we obtained the lowest annual investment costs in Pareto solutions in Table 2.

To verify that the PSO algorithm has obtained the optimal solution, the iterative convergence curve comparison between PSO and GA (Genetic algorithm) is shown in Figure 6. According to the iterative convergence curve of Figure 6, it can conclude that the PSO has fewer convergence times, more effectively avoid local convergence, and has better stability than the GA. We know that the PSO algorithm can obtain the optimal solution.

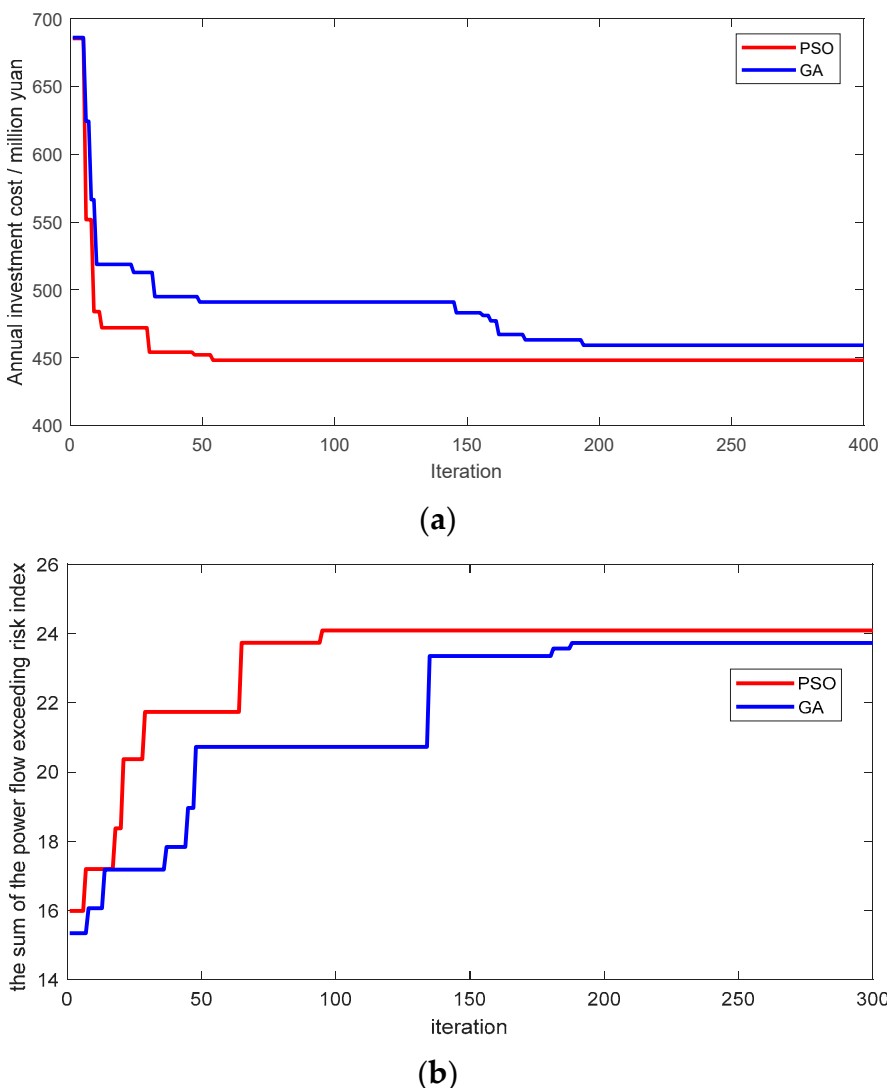

**Figure 6.** The iterative convergence curve. (**a**) Iterative convergence curve of energy storage annual investment cost. (**b**) Iterative convergence curve of the sum of improved power flow exceeding the risk index.

Then, the following three scenarios are compared and analyzed: (1) the BESS is configured in a node in the sensitive and vulnerable line, and 31 nodes are selected in this paper; (2) the BESS is configured in the nodes in the shortest path except for the sensitive and vulnerable lines. This paper selects 27 nodes; (3) 21 nodes are selected for other nodes with BESS configured outside the shortest path. Under the three conditions, each node is connected to 541.15 MW of BESS, and the improved power flow exceeding risk index of sensitive and vulnerable AC lines is shown in Tables 6–8.

**Table 6.** The improved power flow exceeding risk index of the AC line with vulnerability after node 31 is connected to BESS.

| AC Line | Initial Power/p.u. | Power after Fault/p.u. | LODF | The Improved Power Flow Exceeding Risk Index |
|---|---|---|---|---|
| L30-31 | 1.02 | −1.68 | −0.68 | −0.71 |
| L14-19 | −0.34 | −1.3 | −0.41 | −0.83 |
| L15-14 | −0.34 | −1.3 | −0.24 | −0.87 |
| L15-12 | 0.34 | 1.3 | 0.24 | 0.39 |
| L17-16 | 4.79 | 6.37 | 0.39 | 0.51 |
| L16-29 | 2.38 | 6.79 | 1.1 | 0.29 |

**Table 7.** The improved power flow exceeding risk index of the AC line with vulnerability after node 27 is connected to BESS.

| AC Line | Initial Power/p.u. | Power after Fault/p.u. | LODF | The Improved Power Flow Exceeding Risk Index |
|---|---|---|---|---|
| L30-31 | 1.02 | −2.98 | −1 | −0.48 |
| L14-19 | −0.34 | −4.17 | −1.13 | −0.3 |
| L15-14 | −0.34 | −4.17 | −0.96 | −0.22 |
| L15-12 | 0.34 | 4.17 | 0.96 | 0.1 |
| L17-16 | 4.79 | 6.36 | 0.39 | 0.51 |
| L16-29 | 2.38 | 6.69 | 1.08 | 0.3 |

**Table 8.** The improved power flow exceeding risk index of the AC line with vulnerability after node 21 is connected to BESS.

| AC Line | Initial Power/p.u. | Power after Fault/p.u. | LODF | The Improved Power Flow Exceeding Risk Index |
|---|---|---|---|---|
| L30-31 | 1.02 | −2.98 | −1 | −0.48 |
| L14-19 | −0.34 | −1.69 | −0.51 | −0.67 |
| L15-14 | −0.34 | −1.69 | −0.34 | −0.61 |
| L15-12 | 0.34 | 1.69 | 0.34 | 0.27 |
| L17-16 | 4.79 | 6.9 | 0.53 | 0.38 |
| L16-29 | 2.38 | 7.03 | 1.16 | 0.28 |

It can be seen from Tables 6–8 that after the BESS is arranged at the 31 nodes of the sensitive and vulnerable line, the risk index of tidal current out-of-limit of lines L31-30, 19-14, L14-15 and L17-16 exceeds 0.5, and only the risk index of tidal current out-of-limit of lines L15-12 and L16-29 is lower than 0.5; After the BESS is arranged at 27 nodes, only the improve power flow exceeding risk index of line L17-16 exceeds 0.5, and the improve power flow exceeding risk index of other lines does not exceed 0.5; After the BESS is configured at node 21, the improve power flow exceeding risk index of lines 19-14 and L14-15 exceeds 0.5, and the improve power flow exceeding risk index of other lines does not exceed 0.5; It can be seen that the BESS configuration on the sensitive and vulnerable lines has greatly improved the safe operation ability of AC lines.

It can be seen from Figures 7–9 that after the BESS is incorporated into 31 nodes, it is helpful to suppress the oscillation of line power, and the change of line power is lower than 1.5 p.u; after the BESS is incorporated into 27 nodes, the oscillation of line power is increased, and the power variation of some lines is greater than 1.5 p.u; after the BESS is

connected to 21 nodes, the power of AC lines on the shortest path changes irregularly, and the power of some lines increases gradually after 17 s, greatly reducing the system stability. It can be seen that only when the BESS is connected to the sensitive and vulnerable lines that the safe and stable operation capacity of the AC lines can be effectively improved.

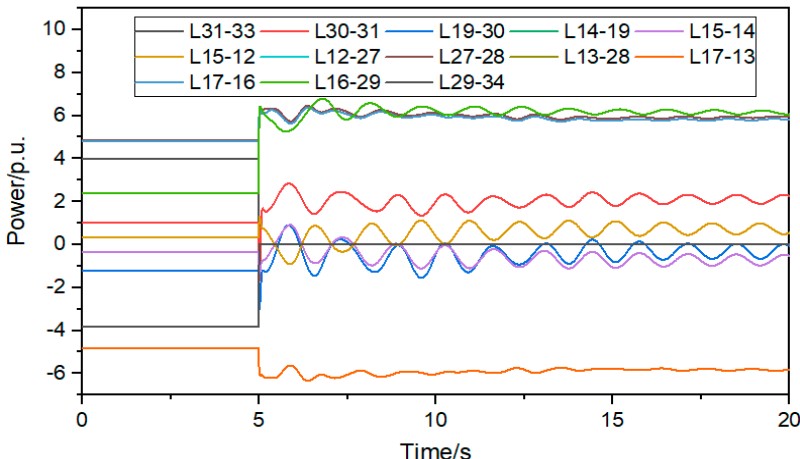

**Figure 7.** The shortest path power change curve after BESS is connected to node 31.

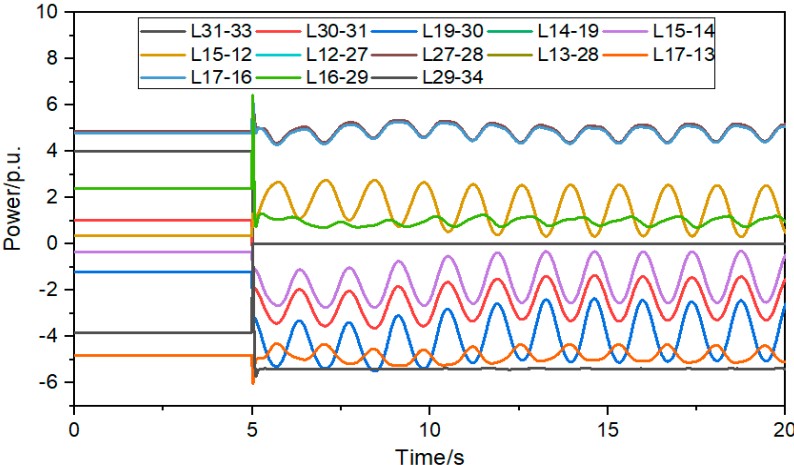

**Figure 8.** Shortest path power change curve after BESS access node 27.

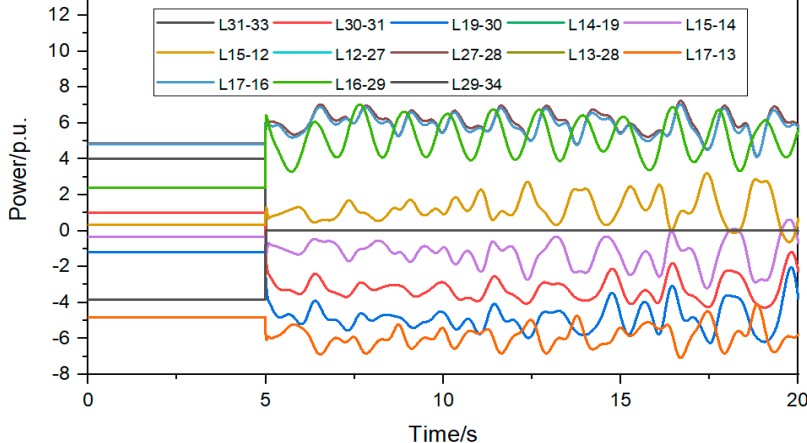

**Figure 9.** The shortest path power change curve after BESS is connected to node 21.

### 5.4. Optimization Configuration Results of Multi-BESS

When the system configures two and three BESS, a set of Pareto solutions is shown in Table 9. The improved power flow exceeding risk index of sensitive and vulnerable AC lines after configurations two and three BESS is shown in Tables 10 and 11. The shortest path power change curve after BESS is shown in Figures 10 and 11.

**Table 9.** The multi-BESS configuration capacity and location.

| Nodes | BESS Power/MW | ESS Capacity/MW·h | BESS Location | Investment and Construction Costs/Million Yuan | the Sum of the Improved Power Flow Exceeding Risk Index |
|---|---|---|---|---|---|
| 2 nodes | 88.09 | 88.09 × 2 | 29 | 248.89 | 17.78 |
| | 299.25 | 299.25 × 2 | 31 | | |
| 3 nodes | 88.09 | 88.09 × 2 | 29 | 248.89 | 17.78 |
| | 11.87 | 11.87 × 2 | 19 | | |
| | 287.38 | 287.38 × 2 | 31 | | |

**Table 10.** The improved power flow exceeding risk index of sensitive and vulnerable AC lines after the configuration of two BESS.

| AC Line | Initial Power/p.u. | Power after Fault/p.u. | LODF | The Improved Power Flow Exceeding Risk Index |
|---|---|---|---|---|
| L30-31 | 1.02 | −2.98 | −1 | −1.38 |
| L14-19 | −0.34 | −4.17 | −1.13 | −1.48 |
| L15-14 | −0.34 | −4.17 | −0.96 | −0.5 |
| L15-12 | 0.34 | 4.17 | 0.96 | 0.5 |
| L17-16 | 4.79 | 6.36 | 0.39 | 0.58 |
| L16-29 | 2.38 | 6.69 | 1.08 | 0.5 |

**Table 11.** The improved power flow exceeding risk index of sensitive and vulnerable AC lines after configuration three BESS.

| AC Line | Initial Power/p.u. | Power after Fault/p.u. | LODF | The Improved Power Flow Exceeding Risk Index |
|---|---|---|---|---|
| L30-31 | 1.02 | −2.98 | −1 | −1.24 |
| L14-19 | −0.34 | −1.69 | −0.51 | −1.48 |
| L15-14 | −0.34 | −1.69 | −0.34 | −0.5 |
| L15-12 | 0.34 | 1.69 | 0.34 | 0.5 |
| L17-16 | 4.79 | 6.9 | 0.53 | 0.58 |
| L16-29 | 2.38 | 7.03 | 1.16 | 0.5 |

It can be seen from Table 6 that when two and three BESS are configured on sensitive lines, the total power of the BESS is the same, and the annual investment cost and the sum of improved power flow exceeding risk index are also the same. Moreover, the sum of the power of 19 nodes and 31 nodes, when three BESS are configured on sensitive lines, is equal to the sum power of configured two BESS. It can be seen from Tables 7 and 8 that the improved power flow exceeding risk index for sensitive and vulnerable AC lines is greater than 0.5, and when three BESS are configured, the impact on the improved power flow exceeding risk index is relatively small. It can be seen from Figures 9 and 10 that when two and three BESS are configured for sensitive lines, the power variation of the line is much

lower than configuring one BESS for sensitive lines. And suppressing power oscillation is greatly improved. We know that configuring BESS for multiple nodes eliminating the impact of DC power transfer on AC lines is better than the BESS configured for a single node. Configuring BESS for multiple nodes to improve the safe and stable operation ability of AC lines is better than the BESS configured for a single node. Secondly, when the configured BESS quantity is greater than 2, it has a small impact on the sum of improved power flow exceeding risk index and annual investment cost.

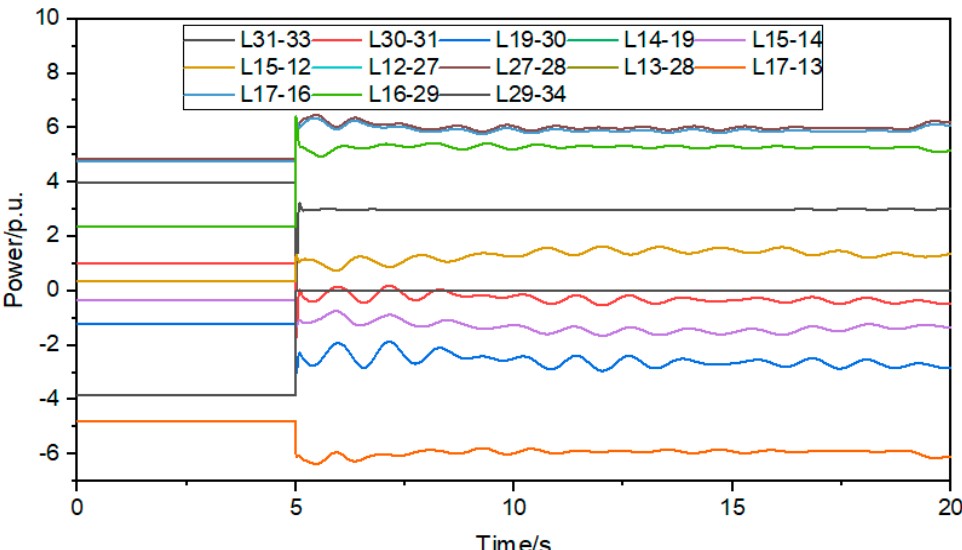

**Figure 10.** Shortest path power change curve after configuring 2 BESS.

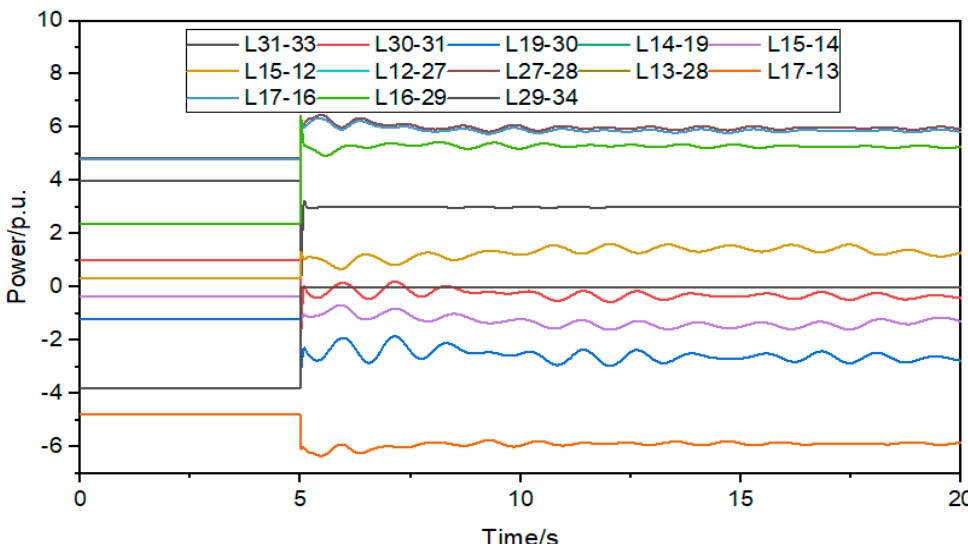

**Figure 11.** Shortest path power change curve after configuring 3 BESS.

## 6. Conclusions

The method of identifying the sensitive and vulnerable transmission lines based on improved power flow exceeding risk index is proposed, and it can apply to AC/DC hybrid systems, weak power grids, and other power systems. And the proposed method can quickly identify the sensitive and vulnerable transmission lines in the shortest path and simplify the method of calculating the branch disconnection coefficient. The method would not have to repeatedly calculate the impedance matrix of the line disconnection and connection.

The optimal energy storage configuration can be obtained by the multi-objective optimal mathematical model, including minimizing the annual investment cost of BESS and maximizing the sum of the improved power flow exceeding the risk index of the sensitive and vulnerable transmission lines. The BESS for sensitive and fragile lines can quickly eliminate the impact of DC power transfer on AC lines, and the BESS can also suppress power fluctuations and greatly improve the transient stability operation ability of the power grid.

The proposed energy storage configuration method does not only improve the transient stability of the power grid. Simultaneously, during the stability power grid, the configured energy storage can actively supports voltage, frequency, etc. The configuration method proposed in this article enriches the application scenarios of energy storage. The safe and stable operation ability of the system is greatly improved.

In the future, we will study the HIL implementation of the proposed system along with the proposed optimal allocation method. After this, the fault disturbance, power system planning BESS resources will be explored through the robust optimization-based model.

**Author Contributions:** Conceptualization, L.J. and T.Z.; investigation, Y.T., L.J., B.Z., X.S. and Y.X.; resources, B.Z., X.S. and Y.X.; data curation, L.J. and T.Z.; writing—original draft preparation, L.J. and T.Z.; writing—review and editing, L.J., T.Z. and S.M.; supervision, T.Z. and S.M.; project administration, Y.T.; funding acquisition, Y.T. All authors have read and agreed to the published version of the manuscript.

**Funding:** This research was funded by the Technology Project of State Grid Sichuan Electric Power Company (No. B1199721009N).

**Data Availability Statement:** The data presented in this study are available on request from the corresponding author.

**Conflicts of Interest:** The authors declare no conflict of interest.

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
