# Peer review of "Optimal Configuration of Battery Energy Storage for AC/DC Hybrid System Based on Improved Power Flow Exceeding Risk Index"

_electronics, doi:10.3390/electronics12143169_

Round 1
Reviewer 1 Report
This paper studies the Optimal configuration of energy storage for AC / DC hybrid system based on power flow exceeding risk index, the idea can be good and the following are my comments for its publication:
1. The introduction is too short, please give more insights into your motivation and background information.
2. The literature review is not enough, please give more comparisons regarding research on energy storage configuration in the energy system. The following can be compared if suitable: Optimal stochastic deployment of heterogeneous energy storage in a residential multienergy microgrid with demand-side management.
3. The language should be checked and improved.
4. All the figures should be improved and plotted better.
5. How do you guarantee the convergence of your PSO method
6. More compassion cases should be done to verify your method's effectiveness
7. The first paragraph of the conclusion is too long
8. Future work should be added
can be improved
Author Response
Dear Reviewer
Thank you very much to the reviewers for their valuable feedback on this article! After receiving the modification comments, we immediately made detailed and in-depth modifications to the entire text. Below, we will respond to the comments made by the four reviewers one by one. We hope that our responses and modifications can satisfy the and reviewers. Our responses to the opinions of the four reviewers are on pages 1-10, pages 11-14, pages 15-22, pages 23-31 of the document, respectively.
Kind regards,
Mr. Jiang

Reviewer 2 Report
Optimal configuration of energy storage for AC / DC hybrid system based on power flow exceeding risk index
Review
In this paper is proposed an optimal configuration of energy storage for AC-DC hybrid systems based on power flow exceeding the risk index, for eliminating the impact of power transfer on transmission lines. Taking into consideration the safe and stable operation of the transmission lines, a multi-objective mathematical model of energy storage configuration for the AC-DC hybrid system is established, to minimize the investment cost of energy storage and to maximize the power flow exceeding the risk index.
The topic is important and of interest for readers. The manuscript is well structured.
However. this reviewer suggests some issues before acceptance:
1. The novelty of the paper should be explained in details.
2. The state of the art must be enriched with more relevant references, published in journals of large circulation, that can be easily retrieved.
3. Regarding the mathematical model presented by equations (1)-(18): it must be explained if it is the original work of the authors, or if it is retrieved from literature. The mentioned in line 117 publication [21] “W. Y. Research online identification and elimination methods of active power flow over-limit of transmission sections in power networks. Jilin University, 2021” cannot be retrieved.
4. The used Power System Analysis Software Package (PSASP) must be identified by using the exact reference, or by an explanation if it is in-house developed software. Same for CEPRI36V7 power grid model of the same software.
5. Why was used the particle swarm algorithm PSO for solving the optimization problem (in Section 4 Model-solving method”)? Please explain.
6. The minimization of investment cost of energy storage, which is mentioned in Abstract, does not exist. The economic estimation of the amounts of money from Table 2 must be analyzed in details.
7. In section 6. Conclusions it is not necessary to repeat what was done in the main body of the manuscript. It is better to focus on results and possible applications.
8. Consider to fragment the too big paragraphs, such as in section 1. Introduction and in section 6. Conclusions.
Moderate editing of English language required.
Author Response

(The authors gave the same response as above.)

Reviewer 3 Report
1. Equation (3) must be explained in more detail. Moreover, the structure of the matrices and vectors referred in equations (3) to (7) must be illustrated.
2. The main problem of this kind of solutions, based metaheuristic approaches, is the lack of information concerning the adjustment of the method parameters. This must be addressed in the paper: in which way was turned each parameter of the PSO algorithm?
3. Another problem concerns the impossibility of ensuring to reach the real optimal solution. Therefore, at least, the authors must to compare their solution with other approaches used in literature for similar problems, e.g. Genetic algorithm, among others. At least other two solutions must be tested.
The language, in general, is correct.
Author Response

(The authors gave the same response as above.)

Reviewer 4 Report
This paper proposes an optimal strategy, for the allocation on energy storage systems in hybrid microgrids, based on the risk index of power flow out-of-limit, in order to eliminate power transfer after DC blocking fault. In general, the paper is of good quality, while the manuscript is well organized and well written. The language level is good. However, there are some key points that the authors have to elaborate more, to effictively highlight the contribution of their work. My comments are summarized below:
(1) The state-of-the-art presentation, carried out in the Introduction Section is limited. Only four scientific literature references are presented and discussed. In addition, the advantages of the proposed method, compared to the presented ones have to be indicated. A more comprehensive literature review has to be performed and the contribution of the proposed method has to be further discussed and elaborated in detail.
(2) The term "energy storage" is very generic. I think that the authors have to provide more details on this. In modern microgrids, various energy storage configurations (hybrid in many cases) are utilized, e.g., supercapacitors, batteries, flywheels etc. Each system presents different features, in terms of dynamic behavior (time constants, transient response), affecting so their ability to support the microgrid in cases of fault. Do these characteristics affect your method? Please discuss on this.
(3) The selection of the PSO algoritm for the optimal selection of the energy storage is not explained. Please provide more details on this. Is the selected algorithm advantageous, among similar methods?
(4) The ESS capacity of the power grid is not given (although discussed). Only the rated power (800W and 1600W). What is the real capacity and ESS type? Does it affect the proposed allocation strategy?
(5) How are the cost coefficients calculated? Are they affected by ESS type? Please elaborate in detail.
(6) As I understand,the studied system is really large and complicated, to validate experimentally. As a fututure outlook, I would suggest the authors the HIL implementation of the proposed system along with the proposed optimal allocation method, in order to obtain a more realistic validation.
There are some minor grammar, typo and syntax mistakes in the manuscript, although the language level is good. Please correct all mistakes and carefully proofread your paper.
Author Response

(The authors gave the same response as above.)

Round 2
Reviewer 2 Report
The authors' responses to my previous comments are missing.
The "Author Response File" contains irrelevant information.
There is no specific response to my previous comments.
None of my suggestions was clearly addressed.
Without a clear response, my recommendation remains the same as before.
Eventually, resubmit the correct file.
Same evaluation as in my previous report.
Author Response
Dear Reviewer
Thank you very much to the reviewers for their valuable feedback on this article! After receiving the modification comments, we immediately made detailed and in-depth modifications to the entire text. Below, we will respond to the comments made by the four reviewers one by one.
Thank you very much for the reviewer's feedback again. We apologize for not responding well to your suggestions or comments in the first modification. Below, we will provide a detailed and serious response to your first comment. We hope that our responses and modifications can satisfy the reviewers.
Our responses to the opinions of the your reviewers are on pages 1-11,
Kind regards,
Mr. Jiang

Reviewer 3 Report
1. The authors have failed in describing the adjustment of the method parameters. The revision only add some references, but the process is not explained. This produces two problems:
a. The use of this method in other systems is not practical, this mainly because the parameters must be turned for the new system, and such a process is not presented.
b. The only way to ensure that the parameters used in references [33]-[34] are the correct ones requires that those references use the same method for the same system, thus no contribution will be presented in this pape.
In conclusion, there is needed a detailed description of the way in which each parameter of the PSO algorithm was turned.
2. Moreover, the authors have not addressed this important remark: “Another problem concerns the impossibility of ensuring to reach the real optimal solution. Therefore, at least, the authors must to compare their solution with other approaches used in literature for similar problems, e.g. Genetic algorithm, among others. At least other two solutions must be tested.”
In conclusion, the authors have not addressed the main concerns pointed out in the previous review round. Therefore, this reviewer finds the paper not suitable for publication.
The language is correct.
Author Response
Dear Reviewer
Thank you very much to the reviewers for their valuable feedback on this article! After receiving the modification comments, we immediately made detailed and in-depth modifications to the entire text. Below, we will respond to the comments made by the reviewers .
thank you very much for the reviewer's feedback on this article, which has deeply inspired the author! We hope that our responses and modifications can satisfy the and reviewers. Our responses to the opinions of the your reviewers are on pages 12-15.
Kind regards,
Mr. Jiang

Reviewer 4 Report
I would like to thank the authors for their answers. All my comments have been succesfully addressed. The manuscript overall quality has been significantly improved.
The languege level has been improved. Please carefully proofread your manuscript, prior to publication.
Author Response
Dear Reviewer
Thank you very much to the reviewers for their valuable feedback on this article! After receiving the modification comments, we immediately made detailed and in-depth modifications to the entire text. We hope that our responses and modifications can satisfy the and reviewers.
Kind regards,
Mr. Jiang

Round 3
Reviewer 2 Report
Following up with previous review and suggestions, the manuscript can be recommended for publication.
Reviewer 3 Report
The authors have finally answered my questions.
The language is almost correct.